# Meta-Learning to Teach Semantic Prompts for Open Domain Generalization in Vision-Language Models

**Shirsha Bose**                                                     *shirshabosecs@gmail.com*
*Technical University of Munich*

**Mainak Singha**                                              *mainaksingha.iitb@gmail.com*
*Indian Institute of Technology Bombay*

**Ankit Jha**                                                        *ankitjha16@gmail.com*
*LNM Institute of Information Technology Jaipur*

**Souradeep Mukhopadhyay**                   *souradeepmukhopadhyay99@gmail.com*
*Indian Institute of Science*

**Biplab Banerjee**                                                  *getbiplab@gmail.com*
*Indian Institute of Technology Bombay*

**Reviewed on OpenReview:** *https: // openreview. net/ forum? id= uJELgNGiMW*

## Abstract

Open Domain Generalization (ODG) addresses the challenges posed by domain and category shifts between labeled training sources and unlabeled target domains. Current state-of-the-art methods struggle with the limitations of traditional CNN backbones, leading to reduced generalization and increased error rates in detecting target open samples without prior knowledge. Additionally, recent CLIP-based prompt learning approaches fail to distinguish between known and unknown classes effectively, resulting in suboptimal performance. To address these challenges, we propose METAPROMPT, which leverages the semantic strengths of the vision-language model CLIP and the "learning-to-learn" capabilities of Meta-Learning to achieve robust generalization across domain and category shifts. Our framework introduces three key innovations: First, we approach ODG as a multi-class classification problem that includes both known and novel categories, designing novel prompts capable of detecting unknown class samples across multiple domains. These prompts are trained using Meta-Learning with momentum updates, enabling smooth and accurate differentiation between known and unknown classes. Second, we introduce a novel domain-agnostic semantic attention-based prompt alongside domain-focused prompts to enhance robustness in classifying unknown classes across various domains. Finally, we incorporate an unsupervised contrastive loss during episodic Meta-Training, which reinforces the boundaries in the metric space between known and unknown classes, thereby enhancing "unknown" class awareness in the prompts. METAPROMPT has demonstrated its superiority through extensive testing on diverse datasets, excelling in both closed and open-set DG scenarios and consistently outperforming existing solutions.

## 1 Introduction

Advancements in deep learning have significantly transformed computer vision tasks such as object detection, segmentation, and classification. However, these models often face challenges with domain shifts, where the

distribution of test data (target domain) differs from the training data (source domain). Techniques such as domain adaptation (DA) and domain generalization (DG) have been developed to address this. Domain adaptation (DA) Li et al. (2021); Baktashmotlagh et al. (2013); Singha et al. (2023) focuses on adapting models to new domains using labeled source data and unlabeled or sparsely labeled target data, employing techniques like feature alignment and adversarial training Ganin & Lempitsky (2015a). On the other hand, domain generalization (DG) Li et al. (2019a; 2020); Wang et al. (2021) aims to learn generalized models from multiple source domains to ensure robust performance on unseen target domains without accessing their data during training. Despite these advancements, traditional DA and DG approaches are limited in recognizing novel category samples. These models typically assume that the number of categories remains constant during both training and inference stages. Consequently, they tend to misclassify novel category samples as known training categories during inference, which is impractical for open-world sample recognition. Open domain generalization (ODG) addresses this issue by considering both known category samples and unique samples within its training framework, making it highly relevant for real-world applications like autonomous driving and remote sensing. However, research targeting ODG remains relatively scarce. Methodologies such as DAML Shu et al. (2021) and MEDIC Wang et al. (2023b) integrate meta-learning techniques to enhance classifier reliability for training classes in the context of ODG. Nonetheless, these approaches encounter challenges when dealing with diverse open-world data and limited adaptability. This paper aims to explore and address these challenges, proposing novel strategies to improve the robustness and adaptability of models in open-world environments.

Vision-language models such as CLIP Radford et al. (2021) and ALIGN Jia et al. (2021) have emerged as robust solutions for various computer vision tasks. CLIP, in particular, leverages vast image and text datasets to create comprehensive embedding spaces through multi-modal contrastive training, excelling in zero-shot and few-shot learning scenarios. Despite its success, CLIP faces challenges in domain generalization due to overfitting and the limitations of traditional prompt tuning, which often fails to enforce cross-domain generalization. This shortcoming underscores the need for more robust prompt learning strategies adaptable to diverse and unknown domains. In recent research on open domain generalization (ODG), most methods focus on either the one-against-all classifier approach or the threshold-dependent approach, which determines whether a sample is out of distribution based on specific thresholding of the probability scores. Even the latest CLIP-based prompt engineering methods, such as CoOp Zhou et al. (2022b), CoCoOp Zhou et al. (2022a), STYLIP Bose et al. (2024), and MaPLe Khattak et al. (2023a), fail to adequately address the ODG problem due to the misalignment of text prompt embeddings with the concept of novel or unknown classes. CLIPN Wang et al. (2023a) introduces a negative concept by introducing 'no' words for text inputs of CLIP that capture the corresponding negation semantics of images for out-of-distribution (OOD) tasks.

To address these limitations, we propose a novel methodology using meta-learning to make the prompt embeddings aware of the textual term 'unknown', enabling them to detect or segregate open samples from closed counterparts. This approach aims to enhance the robustness and adaptability of vision-language models in handling open-domain environments, contributing to the broader field of domain generalization. Our motivation of using CLIP for ODG stems from the need to address these limitations by leveraging its multi-scale features. We aim to generate semantically aware tokens for prompt learning. This facilitates better alignment between image and prompt characteristics to enhance the model's ability to handle open and closed-set classification tasks. To achieve this, we propose METAPROMPT, a model trained in a meta-training setup inspired by model-agnostic meta-learning (MAML) Finn et al. (2017). Inspired by ODG-CLIP Singha et al. (2024), where a pretrained stable-diffusion model is leveraged to generate open-set samples and perform prompt learning to classify novel classes as 'unknown', we focus on robust prompt tuning and improved visual feature extraction in METAPROMPT . This dual focus ensures balanced performance across known and unknown classes, providing a cohesive solution to the challenges of ODG and setting a new benchmark for generalization in unseen and diverse domains. We state our contributions as follows:

- We introduce a novel meta-learning-based prompt learning method, METAPROMPT, for open domain generalization.
- We leverage the multi-scale image features of the pre-trained CLIP to generate the BatchNorm statistics. Additionally, we propose the semantic attention block (SemAtt) that helps generate more semantically meaningful prompt tokens and improves clustering between open-set and closed-set classes.

- Our extensive experiments showcase the efficacy of METAPROMPT for the ODG task on six benchmark datasets. We also surpass the state-of-the-art closed-set domain generalization methods by a significant margin.

## 2 Related Works

### 2.1 Open-Set Recognition (OSR)

The OSR challenge, which aims to effectively identify novel-class samples using only training exemplars from closed-set classes, has become a focal point of research Bendale & Boult (2016); Kong & Ramanan (2021); Pal et al. (2023); Vaze et al. (2021). In this pursuit, generative OSR approaches have emerged as promising strategies. These methods augment the training set with artificially synthesized categories, enhancing the model's ability to generalize and recognize novel-class samples during evaluation Du et al. (2022); Ge et al. (2017); Zhou et al. (2021a). Moreover, recent advancements in vision-language models, such as CLIP, have presented new opportunities for tackling the OSR challenge. CLIPN, for instance, introduces a novel negative prompt learning strategy leveraging CLIP's capabilities Wang et al. (2023a). This approach has demonstrated superior performance compared to existing alternatives Fort et al. (2021); Ming et al. (2022), highlighting the potential of vision-language models in open set recognition tasks.

**Open-Set Domain Adaptation (OSDA):** Open Set Domain Adaptation (OSDA) operates in a transductive cross-domain environment similar to Open Set Recognition (OSR), adopting an OSR-like scenario. Typically, OSDA addresses the presence of out-of-distribution samples by either randomly combining input images or using GAN-based models to generate pseudo-open data Kong & Ramanan (2021); Mancini et al. (2020b). However, these generated images often lie on a lower-dimensional manifold in open space and lack rich semantics. Discriminative models, on the other hand, employ metric learning, reconstruction loss, or the confidence of closed-set classifiers to identify open data Chen et al. (2021); Yoshihashi et al. (2019). CLIPN Wang et al. (2023a) introduced a novel negative prompt learning strategy for OSR using CLIP, significantly outperforming other vision-language model-based methods Fort et al. (2021); Ming et al. (2022). Nonetheless, these models struggle to handle distribution shifts between test and training domains.

**Open Domain Generalization:** Open Domain Generalization (ODG) presents more complex challenges than Open Set Recognition (OSR) and Open Set Domain Adaptation (OSDA) due to its inductive nature and lack of information about the target domain. ODG in supervised learning requires models to generalize across unseen target distributions without prior exposure to the target domain during training, pushing the boundaries of model generalization capabilities. Early domain generalization (DG) research focused on domain adaptation (DA) models to address domain distribution discrepancies. Domain augmentation reduces these discrepancies by generating synthetic domains and integrating them into the existing domain set, enhancing model generalization. The field has advanced with diverse DG strategies, including self-supervised, ensemble, and meta-learning.

The notion of ODG was first introduced in Shu et al. (2021), extending domain-augmented meta-learning. MEDIC Wang et al. (2023b) proposes a comprehensive approach, including domain and class-wise gradient matching, to establish a balanced decision boundary across closed-set classes. ODG-Net Bose et al. (2023) synthesizes images using disentangled style and content information and trains an open-set classifier. Recent studies by Zhu & Li (2021) and Yang et al. (2022) have expanded the ODG framework to include single-source domain scenarios. Recently, Singha et al. (2024) has introduced a new language-guided paradigm to solve the ODG problem in the vision-language modeling (VLM) perspective using an unknown class prompt. This is the first work in ODG with VLM using CLIP, and our proposed method also counterparts the VLM with a meta-learning method. These developments highlight the ongoing evolution of ODG methodologies as researchers strive to improve models' ability to generalize without prior target domain knowledge.

### 2.2 Meta Learning

Meta-learning approaches have been effectively addressing the learning of deep learning models under data-scarce scenarios and can be classified into metric-based methodsVinyals et al. (2017);Snell et al. (2017),

model-based methodsMishra et al. (2018);Graves et al. (2014) and gradient-based learning methodsRavi & Larochelle (2016); Nichol et al. (2018). Model-agnostic meta-learning (MAML)Finn et al. (2017) is the gradient-based learning approach in which the model can adapt to new tasks with minimal fine-tuning after being trained on various tasks. In Domain generalization, meta-learning is efficiently utilized to tackle the domain shifts. Li et al. (2018) proposes MAML-based domain generalization (MLDG), where the model is trained on different source domains and tested on unseen target domains. Zhao et al. (2023), utilized meta prompt learning, where the parameters are regularised after every update to make the model generalizable to unseen domains. Li et al. (2023), GRAM learns prompt initialization and a gradient regulating function for cross-domain generalization.

## 2.3 Vision Language Models and Prompt Learning

Multi-modal vision-language models (VLMs) have significantly advanced image recognition tasks with minimal task-specific fine-tuning. VLMs such as CLIP Radford et al. (2021) and VisualBERT Li et al. (2019b) use advanced language models like BERT Devlin et al. (2018) and GPT Radford et al. (2018) alongside convolutional neural networks and vision transformers Vaswani et al. (2017) for visual analysis. Traditionally, VLMs relied on manually crafted textual prompts, which can be complex and tiresome. Prompt learning methods Zhou et al. (2022b;a); Bulat & Tzimiropoulos (2023); Khattak et al. (2023a) have gained traction for tailoring prompts effectively for downstream tasks by treating token embeddings as learnable variables constrained by image features.

Recently, CLIP has been employed to tackle challenges in domain adaptation (DA) Singha et al. (2023) and domain generalization (DG) Zhang et al. (2021); Bose et al. (2024). However, CLIP's text encoder, trained with positive instances, fails to recognize instances with negation. To address this, CLIPN Wang et al. (2023a) introduces a "no-text encoder" that assimilates out-of-distribution (OOD) samples. ODG-CLIP Singha et al. (2024) assigns OOD samples from diverse unseen domains to the "unknown" class and applies prompt learning, generating open-set samples using stable-diffusion controlled by positive and negative prompts. *In our proposed METAPROMPT , we aim to perform open-set DG with meta-learning, focusing on learning domain-agnostic features for prompt learning.*

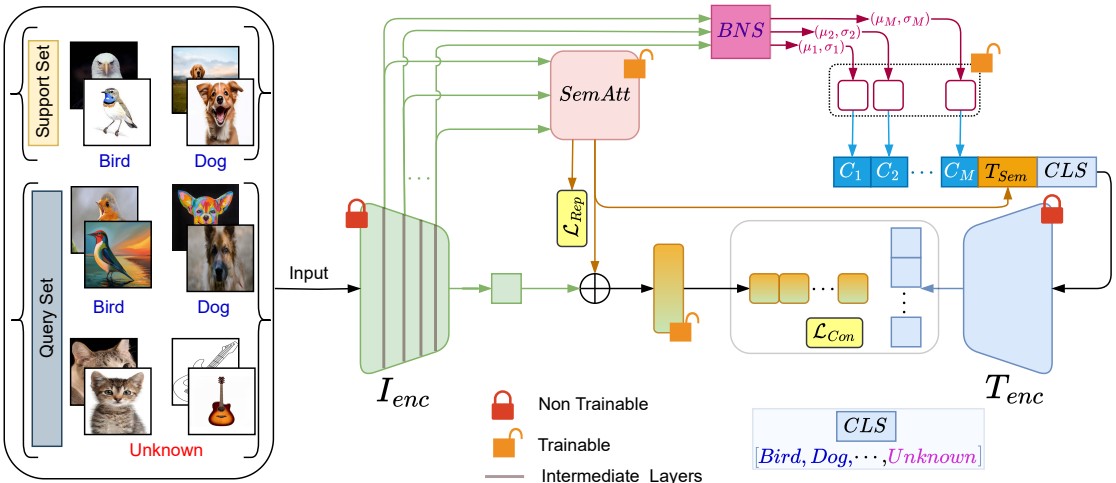

Figure 1: The architecture of our proposed METAPROMPT , where $I_{enc}$ and $T_{enc}$ denote the image and text encoders from the frozen pre-trained CLIP, respectively. We extract the BatchNorm features from the BatchNorm Statistics block ($BNS$). Also, we perform the prompt learning using the context tokens $C_M$ and the semantic token $T_{Sem}$ produced from the semantic attention block ($SemAtt$, shown in Fig 2).

## 3 Methodology

This section first discusses the mathematical notation and goals of open domain generalization (ODG). Afterwards, we explain our proposed methodology and network optimization in Sections 3.2 and 3.3, respectively.

### 3.1 Preliminaries

ODG aims to develop a model that can generalize to a new or unseen target domain using data from $M$ different source domains. Let $\mathcal{X}$ denote the input space and $\mathcal{Y}$ represent the label space. We denote the samples from the source domains as $\{\mathcal{D}^{\mathcal{S}^i}\}_{i=1}^M$, where $\mathcal{D}^{\mathcal{S}^i} = \{(x_j^{\mathcal{S}^i}, y_j^{\mathcal{S}^i})\}_{j=1}^{n_i} \sim p_{\mathcal{X}\mathcal{Y}}^{\mathcal{S}^i}$. Here, $x \in \mathcal{X} \subseteq \mathbb{R}^d$ represents an input image, $y \in \mathcal{Y}$ denotes the class associated with $x$, and $p_{\mathcal{X}\mathcal{Y}}^{\mathcal{S}^i}$ denotes the joint distribution in the $i$-th source domain. Each source domain $\mathcal{D}^{\mathcal{S}^i}$ is partitioned into a support set $\mathcal{D}_{sup}^{\mathcal{S}^i}$ and a query set $\mathcal{D}_{que}^{\mathcal{S}^i}$. The support set $\mathcal{D}_{sup}^{\mathcal{S}^i}$ is used for learning the model parameters: $\mathcal{D}_{sup}^{\mathcal{S}^i} = \{(x_{j_{sup}}^{\mathcal{S}^i}, y_{j_{sup}}^{\mathcal{S}^i})\}_{j=1}^{n^{\mathcal{S}^i}}$. The query set $\mathcal{D}_{que}^{\mathcal{S}^i}$ is used for evaluating the model's performance: $\mathcal{D}_{que}^{\mathcal{S}^i} = \{(x_{j_{que}}^{\mathcal{S}^i}, y_{j_{que}}^{\mathcal{S}^i})\}_{j=1}^{n^{\mathcal{S}^i}}$. The goal is to learn a model $f : \mathcal{X} \to \mathcal{Y}$ that performs well on a new or unseen target domain $\mathcal{D}^{\mathcal{T}} = \{(x_j^{\mathcal{T}}, y_j^{\mathcal{T}})\}_{j=1}^{n_{\mathcal{T}}}$ with joint distribution denoted as $p_{\mathcal{X}\mathcal{Y}}^{\mathcal{T}}$. Also, source and target domains follow different distributions such as $p_{\mathcal{X}\mathcal{Y}}^{\mathcal{S}} \neq p_{\mathcal{X}\mathcal{Y}}^{\mathcal{T}}$. The key challenge in training our METAPROMPT is that $p_{\mathcal{X}\mathcal{Y}}^{\mathcal{T}}$ is unknown.

### 3.2 Proposed Methodology

In Figure 1, we present an overview of our proposed METAPROMPT . We utilize the frozen pre-trained encoders of CLIP Radford et al. (2021). Specifically, the image and text encoders are denoted as $I_{enc}$ and $T_{enc}$, respectively. Motivated by the success of CLIP in domain generalization Zhang et al. (2021); Bose et al. (2024) and domain adaptation Ge et al. (2022); Singha et al. (2023) tasks, our METAPROMPT framework is designed to train a few learnable layers on top of the pre-trained CLIP encoders for the detection of out-of-distribution (OOD) samples under varying target distribution.

**Image and text encoding:** For input images $x$, the pre-trained CLIP's image encoder $I_{enc}$ provides the latent image features $I_{enc}(x)$. Additionally, we engineered the pre-trained CLIP image encoder to extract intermediate features from its layers, denoted as $\{I_{enc}\}_{l=1}^L$ and pass through the Semantic Attention block $Sem_{Att}$. We then concatenate both the latent image features and output of $Sem_{Att}$ through operator $\oplus$, which are subsequently passed through a learnable linear layer, denoted as $G$, for contrastive training with the text features to achieve domain generalization.

The text encoder of CLIP comprises transformer layers, as proposed in Vaswani et al. (2017), which process the image-conditioned features, as depicted in Figure 1. The text features are obtained from the pre-trained text encoder of CLIP by passing $M$ context tokens $[C_1, C_2, \cdots, C_M]$ along with the semantic token $T_{sem}$ and the class tokens $[CLS]$. Precisely, $T_{enc}$ processes the input vector collectively represented as $[C_1, C_2, \cdots, C_M][T_{Sem}][CLS]$, conditioned on the corresponding input image feature $I_{enc}(x)$. As discussed, we utilize the intermediate layers of the pre-trained image encoder $I_{enc}$, generating attentive features, which we further leverage for prompt learning, as detailed in the subsequent paragraphs.

**Domain Agnostic Semantic Attention** $Sem_{Att}$**:** To learn more abstract domain agnostic semantic features, we propose the semantic attention block, denoted as $Sem_{Att}$ in Figure 1, which takes the intermediate features of the pre-trained image encoder of CLIP. The detailed understanding of our proposed domain agnostic semantic attention block is shown in Figure 2, which consists of upsampling layers for dimension matching between the features of intermediate layers of pre-trained CLIP image encoder $I_{enc}$ followed by the convolutional layer. After that, we use flatted the obtained convolution maps via global average pooling layer $GAP$, which finally served at three instances, i.e., final visual latent space, semantic token $T_{sem}$ for prompt learning, and discovery loss $\mathcal{L}_{Rep}$. We define the semantic token $T_{sem}$ in Equation 1, where it takes the features obtained from the $GAP$ layer and passes through a single linear layer $\Omega$. It is essential to note that the multi-scale features from the intermediate layers of the image encoder of pre-trained CLIP $I_{enc}$ played a crucial role in dismantling the domain agnostic properties across the seen and unseen classes of the

target domains.

$$T_{sem} = \Omega\Big(Sem_{Att}(\{I_{enc}\}_{i=1}^{L}(x))\Big) \tag{1}$$

**Style-guided Prompt Learning:** Here, we discuss the generation of prompt tokens conditioned to the input image features. We divide our prompt generation method into two steps: i) incorporating BatchNorm (BN) features, i.e., style and content information from the input image $x$, and ii) adding of semantic token $T_{sem}$ for the final prompt generation. As shown in Figure 1, the BatchNorm statistics ($BNS$) module extracts the $(\mu_M, \sigma_M)$ corresponding to multi-scale features from $L$ intermediate layers of CLIP's image encoder $I_{enc}$ with $M \leq L$. Thereafter, we pass these multi-scale BN features to the learnable layers of the meta networks $\phi_{i=1}^{M}$ for generating the $M$ context tokens $[C_1, C_2, \cdots, C_M]$, mathematically defined in Equation 2.

$$C_m = \phi\Big(BNS(\{I_{enc}\}_{i=1}^{L}(x))\Big)_{j=1}^{M} = \phi_j(\mu_i, \sigma_i)_{i,j=1}^{L,M} \tag{2}$$

where $m \in \{1, M\}$. These style-guided context tokens $C_M$s are further appended with the attention-based semantic token $T_{sem}$ obtained from the $Sem_{Att}$ block, defined in Equation 1 and class tokens $[CLS]$. On accumulating these tokens, we pass them through the frozen text encoder of pre-trained CLIP $T_{enc}$, which generates the text encoding for contrastive learning with the image encoding $P(x)$. The generated text encoding learns the prompts concerning the known classes and open-set unknown samples, which we called the new class "`unknown`".

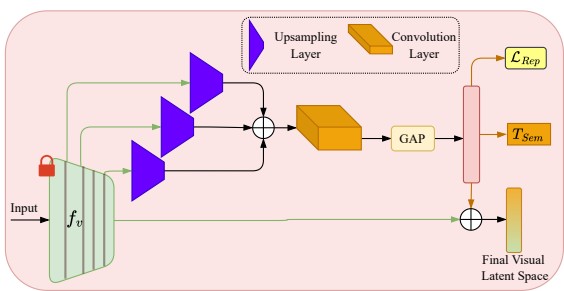

Figure 2: Detailed on the semantic attention block $SemAtt$, which generates the semantic token $T_{sem}$ from the intermediate layers of pre-trained CLIP's image encoder, $I_{enc}$. $\oplus$ and GAP denote the concatenation operator and global average pooling, respectively.

**Algorithm 1:** Pseudocode for METAPROMPT

1: **Input:** $\theta$, $\theta_{sup}$, $\theta_{que}$, $\alpha$, $\alpha_1$, $\alpha_2$
2: **Output:** Trained Model weights $\theta$
3: **while** no converge **do**
4:     **Sample:** $\mathcal{D}_{sup}^{\mathcal{S}}$ and $\mathcal{D}_{que}^{\mathcal{S}}$ sets from $\mathcal{D}^{\mathcal{S}}$
5:     **for** $n$ episodes **do**
6:       $pred_{sup} = \theta(\mathcal{D}_{sup}^{\mathcal{S}})$;
7:       $\theta_{sup} \leftarrow \theta - \alpha_1 \times \nabla_\theta L_1(pred_{sup}, GT_{sup})$;
8:       $pred_{que} = \theta_{sup}(\mathcal{D}_{sup}^{\mathcal{S}})$;
9:       $\theta_{que} \leftarrow \theta - \alpha_2 \times \nabla_{\theta|\theta_{sup}} L_2(pred_{que}, GT_{que})$;
10:     **end for**
11:     Momentum Update: $\theta \leftarrow \alpha\theta_{sup} + (1-\alpha)\theta_{que}$;
12: **end while**
13: **return** $\theta$

## 3.3 Meta-learning for ODG

In this section, we discuss the heuristic approach to tackle the Open Domain Generalization (ODG) problem using a meta-learning setup. We define the optimization functions, i.e., the contrastive loss $\mathcal{L}_{Con}$ and the representation loss $\mathcal{L}_{Rep}$, which are integral to our proposed METAPROMPT framework.

### 3.3.1 Episodic Learning

Episodic learning forms the core of the meta-learning process, where the model is trained on multiple tasks or episodes. Each episode is composed of a support set and a query set. The support set provides the model with examples of tasks, while the query set is used to evaluate the model's generalization capability. By alternating between these sets during training, the model learns to generalize across various tasks, enhancing its ability to deal with unseen classes in ODG.

### 3.3.2 Functioning of Support and Query Sets in Training MetaPrompt

In the METAPROMPT framework, training is based on the concept of episodic learning, where each episode mimics a task the model may encounter during testing. An episode consists of two key components: a *support set* and a *query set*. The support set serves as a small labeled dataset representing a specific task, allowing the model to learn underlying patterns and relationships between input images and their corresponding labels during each episode. This training helps the model adapt to new tasks by generalizing from the few examples provided, a capability crucial for open-domain generalization (ODG), where handling unseen or unknown classes effectively is essential. Meanwhile, the query set evaluates the model's performance within the same episode by presenting samples not seen during the current training phase. This evaluation assesses the model's ability to generalize from the support set to new, unseen instances—an ability critical in ODG scenarios. By alternating between learning from the support set and being evaluated on the query set, the model improves its generalization capabilities across different tasks, better equipping it to handle novel or unseen data during actual deployment.

### 3.3.3 Loss Functions

**Contrastive Loss ($\mathcal{L}_{Con}$):** Contrastive loss aligns text and image embeddings within a shared latent space, which is crucial for tasks involving both modalities. It ensures that embeddings of similar pairs (e.g., text and image from the same class) are close together, while embeddings of dissimilar pairs are far apart. In METAPROMPT, this loss is applied to both the support and query sets. The goal is to minimize the distance between embeddings of matching pairs and maximize the distance between non-matching pairs. This encourages the model to develop a robust embedding space that effectively generalizes across different tasks and instances.

$$\mathcal{L}_{Con} = \min_{\phi,\Omega,P,\theta} \left( \mathbb{E}_{(x,y)\in p(\mathcal{D}^{\mathcal{S}}_{sup})} -\log p(y|x) + \mathbb{E}_{(x,y)\in p(\mathcal{D}^{\mathcal{S}}_{que})} -\log p(y|x) \right) \tag{3}$$

where $p(y|x)$ is defined in Equation 4 with $< \, . \,>$ representing the cosine similarity function and $\tau$ as a temperature hyper-parameter.

$$p(y|x) = \frac{\exp^{(<(T_{enc}(\phi(I_{enc}(x)))),I_{enc}(x))/\tau>)}}{\sum\limits_{y\in\mathcal{Y}} \exp^{(<(T_{enc}(\phi(I_{enc}(x)))),I_{enc}(x))/\tau>)}} \tag{4}$$

**Representation Loss ($\mathcal{L}_{Rep}$):** The representation loss focuses on enhancing the separability between known and unknown classes. It uses an unsupervised contrastive approach to ensure that samples from the same class are clustered together while those from different classes are pushed apart in the latent space. This loss is particularly important in ODG settings, where the model must distinguish between known classes (which it has seen during training) and open-set classes (which are entirely new). By enforcing clear boundaries in the latent space, the representation loss helps the model avoid confusion between these classes.

$$\mathcal{L}_{Rep} = \frac{1}{|A_{sup}|}\sum_{(i,j\in A_{sup})}^{A_{sup}} l_{i,j} + \frac{1}{|A_{que}|}\sum_{(i,j\in A_{que})}^{A_{que}} l_{i,j} \tag{5}$$

where $l_{i,j} = -\log \frac{e^{(<z_i,z_j>)\tau}}{\sum_{k=1}^{N+M} \mathbb{1}_{[k\neq i]} e^{(<z_i,z_k>)\tau}}$ and $< \cdot >$ denotes the cosine similarity function. Here, $z$, $A_{sup}$ and $A_{que}$ represent the embedding space, positive pairs in the support set and positive pairs between the support and query sets, respectively. The representation loss $\mathcal{L}_{Rep}$, by enforcing similar samples to cluster together while keeping dissimilar samples apart, further refines the model's ability to distinguish between known and unknown classes.

**Total Loss $\mathcal{L}_{Total}$:** The total loss function used to train the learnable layers of our proposed METAPROMPT is a combination of the contrastive loss and the representation loss, as defined in Equation 6.

$$\mathcal{L}_{Total} = \mathcal{L}_{Con} + \lambda\mathcal{L}_{Rep} \tag{6}$$

where $\lambda$ is a hyper-parameter that controls the trade-off between the contrastive and representation losses. By minimizing $\mathcal{L}_{Total}$, the model is trained to not only align the text and image embeddings in the latent space but also to effectively separate known and open-set samples, thereby enhancing its generalization capability in open-domain scenarios.

In Section 3.2, we provide a detailed explanation of the meta-training procedure, along with the training algorithm of our proposed METAPROMPT, which illustrates how these loss functions are utilized during the training process.

## 4 Experimental Evaluations

**Datasets:** We evaluate our proposed METAPROMPT with six benchmark datasets for the open-domain generalization, PACS Li et al. (2017b), VLCS Li et al. (2017c), Office-Home Venkateswara et al. (2017), Digits-DG Zhou et al. (2020), Multi-Dataset Shu et al. (2021) and Mini-DomainNet Peng et al. (2019). **Office-Home** Venkateswara et al. (2017) have 15,500 images from 65 classes across four domains, i.e., Art, Clipart, Product, and Real. **PACS** Li et al. (2017a) contains $\approx$ 10K images of seven classes spanning four domains: Artpaint, Cartoon, Sketch, and Photo. **VLCS** Li et al. (2017c) combines images from PASCAL VOC 2007 Everingham et al. (2010), Caltech Fei-Fei et al. (2004), LabelMe Russell et al. (2008), and SUN Xiao et al. (2010), featuring with five classes. **Digits-DG** Zhou et al. (2020) consists of handwritten digit datasets MNIST LeCun et al. (1998), MNIST-M Ganin & Lempitsky (2015b), SVHN Netzer et al. (2011), and SYN Ganin & Lempitsky (2015b). **Multi-Dataset** Shu et al. (2021) is made up of Office-31, STL-10, Visda2017, and four domains from the DomainNet dataset. Peng et al. (2019): Includes 600K images across six domains—real, painting, clipart, quickdraw, infograph, and sketch—divided into 345 classes, with each domain containing 48K to 172K images.

**Architecture Details:** We present the architecture overview of our proposed METAPROMPT in Figure 1. The image encoder is based on ResNet-50, and the text encoder utilizes a transformer model. The prompts are learned through a learnable projector, which inputs the BatchNorm features $(\mu, \sigma)$ from intermediate layers of the frozen CLIP image encoder. This learnable projector consists of a single linear layer followed by a ReLU activation layer to generate context tokens, which are then processed by the text encoder to produce the final text embeddings. In addition to the generation of image conditional context tokens $[C_1, C_2, \cdots, C_M]$, we incorporate semantic information of the given input image captured through the *SemAtt* module, which generates the semantic token $T_{sem}$. The *SemAtt* module uses multi-scale features from the CLIP image encoder (ResNet-50), downsamples them to match the spatial dimensions of the last ResNet-50 layer, and concatenates them, as shown in Figure 2. This concatenated output is processed by a single convolution layer, a ReLU activation layer, and global average pooling (GAP). The GAP layer's output is then passed through a multi-layer perceptron (MLP) to generate the feature embedding corresponding to the input image. This MLP output generates the semantic token $T_{sem}$ through another learnable single-layer MLP for unsupervised representation loss. It creates the final visual latent space by elementwise addition with the ResNet-50 latent space. Including the semantic token $T_{sem}$ leverages global semantic content information for contrastive learning with the final latent visual embedding.

### 4.1 Meta-training step

We adopt the MAML-based meta-training strategy Finn et al. (2017) for the overall training of our proposed METAPROMPT model. As described in Section 3.2, the overall parameters of METAPROMPT are denoted as $\theta \leftarrow \alpha\theta_{sup} + (1 - \alpha)\theta_{que}$, where $\theta_s$ and $\theta_q$ represent the parameters associated with the training models for the support and query sets, respectively. $\alpha$, represents the varying parameter that determines the shift of $\theta$ towards $\theta_{sup}$ or $\theta_{que}$ during the momentum update. During the optimization of METAPROMPT for each episode, we randomly sample six classes with $k$ samples per class, four of which come from the known classes, and two are treated as pseudo unknown classes, referred to as 'unknown'. For $n$ episodes, we

optimize METAPROMPT following the training steps outlined in Algorithm 1. For our experiments we have kept $\alpha$ as 0.5.

The parameter update for $\theta_q$ involves training prompt tokens to make the pre-trained CLIP model aware of the concept of 'unknown', facilitating the detection of outliers or open-set samples during inference. Meanwhile, the parameter update for $\theta_s$ adjusts the prompt for the seen classes during meta-training. The overall model parameters $\theta$ are updated as a momentum update of $\theta_s$ and $\theta_q$. Consequently, our model parameters $\theta$ converge, enabling the model to perform domain generalization in the presence of outliers effectively. The values of $\alpha_1$, $\alpha_2$ (learning rates defined in Algorithm 1) are set as $10^{-2}$, and $10^{-2}$ respectively. Additionally, we set the value of $\lambda$ (defined in Equation 6) to 0.5. The number of context tokens $M$ is set to four. All experiments are conducted with three different seeds, and the results are reported.

## 4.2 Comparison to the Literature

This section compares our METAPROMPT with the benchmark baselines for both the open-set DG and closed-set DG. In the case of open-set DG, we consider three types of state-of-the-art baseline methods and compare the performance of our proposed METAPROMPT with them. In type-I, we denote the performances of conventional CNN-based (ResNet-18 specifically) ODG methods such as Cumix Mancini et al. (2020a), MixStyle Zhou et al. (2021b), DAML Shu et al. (2021), ODG-Net Bose et al. (2023), and MEDIC Wang et al. (2023b). In type-II, we evaluate the efficacy of our METAPROMPT against zero-shot CLIP combined with state-of-the-art ODG methods like OpenMax Bendale & Boult (2016) and OSDA Panareda Busto & Gall (2017). Finally, in type-III, we compare our METAPROMPT with CLIP-based prompt learning methods such as CoOp Zhou et al. (2022b), CoCoOp Zhou et al. (2022a), MaPLe Khattak et al. (2023a), LASP Bulat & Tzimiropoulos (2023), STYLIP Bose et al. (2024), CLIPN Wang et al. (2023a), and ODG-CLIP Singha et al. (2024). To classify the OOD samples, we employ a confidence-driven approach with CoCoOp, MaPLe, PromptSRC, and STYLIP, whereas CLIPN uses a learnable "no" text encoder to identify OOD instances. In contrast, ODG-CLIP considers all novel class samples as a single 'unknown' class.

Additionally, we perform experiments in a closed-set setting using the leave-one-out domain protocol, i.e., $M - 1$ domains for training and $M^{th}$ domain for testing. We compare our METAPROMPT against non-CLIP-based methods such as SWAD Cha et al. (2021b), EoA Arpit et al. (2022), DandelionNet Hu et al. (2023), and ODG-Net Bose et al. (2023). We also compare our METAPROMPT with zero-shot CLIP and other prompt learning-based approaches.

Table 1: Comparative analysis for PACS, VLCS, Office-Home, Digits-DG, Multi-Dataset, and Mini-DomainNet in ODG setting on average ACC and HM over all the domain combinations following leave-one-domain-out protocol. We report the best results in **bold**.

| | Methods | Venue | PACS | | VLCS | | OfficeHome | | Digits-DG | | Multi-Dataset | | Mini-DomainNet | | Average | |
|---|---|---|---|---|---|---|---|---|---|---|---|---|---|---|---|---|
| | | | Acc | HM | Acc | HM | Acc | HM | Acc | HM | Acc | HM | Acc | HM | Acc | HM |
| CNN | Cumix Mancini et al. (2020a) | ECCV'20 | 57.85 | 41.05 | 52.46 | 50.11 | 51.67 | 49.40 | 58.13 | 54.20 | 42.18 | 46.91 | 50.27 | 39.16 | 52.09 | 46.81 |
| | MixStyle Zhou et al. (2021b) | ICLR'21 | 63.35 | 48.30 | 52.30 | 50.61 | 53.52 | 49.53 | 60.23 | 56.35 | 42.18 | 46.91 | 50.43 | 40.25 | 53.67 | 48.66 |
| | DAML Shu et al. (2021) | CVPR'21 | 65.49 | 51.88 | 53.53 | 51.59 | 56.45 | 53.34 | 59.51 | 55.61 | 46.61 | 51.71 | 52.81 | 43.63 | 55.73 | 51.29 |
| | ODG-Net Bose et al. (2023) | TMLR'23 | 68.80 | 55.81 | 57.24 | 56.09 | 59.40 | 56.69 | 63.85 | 60.98 | 49.48 | 54.56 | - | - | 59.75 | 56.83 |
| | MEDIC Wang et al. (2023b) | ICCV'23 | 89.81 | 83.03 | 57.28 | 55.73 | 60.26 | 57.91 | 83.28 | 66.30 | 50.74 | 53.13 | 55.29 | 45.71 | 66.11 | 60.30 |
| CLIP | CLIP Radford et al. (2021) | - | 95.16 | 76.77 | 91.84 | 72.94 | 81.43 | 63.62 | 77.08 | 61.95 | 77.88 | 72.19 | 84.50 | 68.94 | 84.65 | 69.40 |
| | CLIP + OpenMax Bendale & Boult (2016) | - | 93.45 | 79.13 | 92.09 | 73.67 | 81.00 | 61.54 | 76.93 | 62.78 | 78.34 | 73.26 | 81.89 | 69.40 | 83.95 | 69.96 |
| | CLIP + OSDA Panareda Busto & Gall (2017) | - | 92.62 | 75.40 | 90.21 | 70.89 | 82.58 | 67.35 | 80.53 | 65.70 | 74.45 | 75.22 | 82.00 | 73.62 | 83.73 | 71.36 |
| | CoOp Zhou et al. (2022b) | IJCV'22 | 78.77 | 26.87 | 92.02 | 39.26 | 73.85 | 36.26 | 58.54 | 34.81 | 66.03 | 44.61 | 61.13 | 68.34 | 71.72 | 41.65 |
| | CoCoOp Zhou et al. (2022a) | CVPR'22 | 85.76 | 32.93 | 89.47 | 37.01 | 75.38 | 34.38 | 52.77 | 33.50 | 64.84 | 47.57 | 60.63 | 56.30 | 71.48 | 40.28 |
| | MaPLe Khattak et al. (2023a) | CVPR'23 | 93.97 | 48.47 | 89.70 | 43.33 | 79.47 | 33.06 | 70.54 | 43.83 | 69.34 | 62.20 | 74.67 | 60.57 | 79.62 | 48.58 |
| | LASP Bulat & Tzimiropoulos (2023) | CVPR'23 | 88.45 | 30.37 | 90.67 | 39.41 | 76.13 | 34.52 | 60.89 | 35.23 | 66.78 | 50.22 | 62.34 | 61.56 | 74.21 | 41.89 |
| | PromptSRC Khattak et al. (2023b) | ICCV'23 | 94.53 | 43.32 | 90.13 | 42.78 | 80.21 | 36.40 | 75.34 | 44.25 | 65.51 | 59.45 | 73.60 | 62.56 | 79.89 | 48.13 |
| | STYLIP Bose et al. (2024) | WACV'23 | 95.36 | 50.74 | 90.75 | 65.66 | 84.73 | 60.97 | 80.59 | 58.15 | 79.88 | 71.99 | 80.22 | 69.11 | 85.26 | 62.77 |
| | CLIPN Wang et al. (2023a) | ICCV'23 | 96.24 | 45.00 | 84.82 | 50.72 | 84.55 | 42.83 | 81.70 | 45.56 | 77.16 | 62.60 | 77.38 | 66.92 | 83.64 | 52.27 |
| | ODG-CLIP Singha et al. (2024) | CVPR'24 | **99.53** | 99.70 | 95.71 | 86.53 | **98.32** | 96.08 | 91.53 | 78.27 | 84.60 | 90.00 | **95.68** | 94.48 | 94.23 | 90.84 |
| | Meta-MaPLe | - | 92.35 | 84.44 | 89.42 | 73.56 | 86.33 | 78.61 | 91.23 | 71.69 | 81.79 | 78.75 | 79.48 | 69.81 | 86.77 | 76.14 |
| | Meta-STYLIP | - | 95.67 | 88.39 | 93.25 | 78.87 | 91.57 | 87.35 | 92.18 | 72.73 | 84.23 | 83.44 | 86.77 | 78.08 | 90.61 | 81.47 |
| | Meta-ODG-CLIP | - | 99.78 | 99.95 | 96.52 | 88.44 | 98.66 | 98.29 | 92.94 | 80.16 | 86.51 | 92.33 | 96.43 | 95.58 | 95.14 | 92.45 |
| | METAPROMPT + Stable Diffusion | - | 99.13 | 99.95 | 96.53 | 92.85 | 97.58 | 98.89 | 95.27 | 83.69 | 88.65 | 94.14 | 96.19 | 98.79 | 95.56 | 94.72 |
| | METAPROMPT (ours) | - | 98.87 | **99.89** | 95.94 | **89.79** | 96.83 | **97.57** | 93.15 | 80.06 | 86.63 | **91.76** | 94.75 | **96.52** | 94.36 | **92.60** |

**Open-set DG**: In Table 1, we present the performance comparison of our METAPROMPT against state-of-the-art methods for classifying seen and unknown classes in domain generalization. We find that METAPROMPT shows significant improvement over non-CLIP based methods in terms of accuracy and

Table 2: Closed-set domain generalization (leave-one-domain-out) performance on the PACS, VLCS, Office-Home, Digits-DG, and Mini-DomainNet datasets. We report the best results in **bold**.

| | Methods | Venue | PACS | VLCS | OfficeHome | Digit-DG | MiniDomainNet | Average |
|---|---|---|---|---|---|---|---|---|
| CNN | SWAD Cha et al. (2021a) | NeurIPS'21 | 88.10 | 79.10 | 70.60 | - | - | 79.27 |
| | EoA Arpit et al. (2022) | NeurIPS'22 | 88.60 | 79.10 | 72.50 | - | - | 80.07 |
| | DandelionNet Hu et al. (2023) | ICCV'23 | 89.20 | 81.60 | 70.40 | - | - | 80.40 |
| | ODG-Net Bose et al. (2023) | TMLR'23 | 90.66 | 79.85 | 72.92 | 86.75 | 50.16 | 76.07 |
| CLIP | CLIP Radford et al. (2021) | - | 94.89 | 82.14 | 78.40 | 64.59 | 78.73 | 79.75 |
| | CoOp Zhou et al. (2022b) | IJCV'22 | 97.11 | 83.34 | 81.33 | 77.11 | 72.30 | 82.23 |
| | CoCoOp Zhou et al. (2022a) | CVPR'22 | 96.54 | 85.02 | 81.05 | 79.36 | 71.51 | 82.70 |
| | MaPLe Khattak et al. (2023a) | CVPR'23 | 97.72 | 86.75 | 83.52 | 80.25 | 73.87 | 84.42 |
| | LASP Bulat & Tzimiropoulos (2023) | ICCV'23 | 97.02 | 87.25 | 84.13 | 79.92 | 70.67 | 83.80 |
| | StyLIP Bose et al. (2024) | WACV'24 | 98.17 | 87.21 | 85.94 | 81.62 | 80.43 | 86.67 |
| | PromptSRC Khattak et al. (2023b) | ICCV'23 | 98.02 | 86.34 | 83.89 | 82.40 | 76.10 | 85.35 |
| | ODG-CLIP Singha et al. (2024) | CVPR'24 | **99.83** | 95.74 | 96.91 | 96.38 | 96.65 | 97.10 |
| | METAPROMPT (ours) | - | 99.17 | **95.92** | **97.35** | **96.56** | **97.20** | **97.24** |

HM scores at least by 28.25% and 32.30% respectively. Built on top of pre-trained CLIP, we evaluate METAPROMPT against zero-shot CLIP, CLIP + OpenMax, and CLIP + OSDA, noting a rise up in accuracy and HM scores at least of 9.71% and 21.24%. In comparison to recent prompt learning approaches, i.e., CoOp, CoCoOp, MaPLe, LASP, and PromptSRC and STYLIP, which is not mainly based on ODG solutions, our METAPROMPT outperforms them by at least 29.83% in terms of a harmonic score of closed and open-set recognition. The most closely related methods to our METAPROMPT are CLIPN and ODG-CLIP. CLIPN uses a trainable "no" text encoder to distinguish outliers from seen samples. However, it does not directly focus on the ODG problem, and ODG-CLIP is the first CLIP-based VLM, specifically for ODG, solved by using an unknown class prompt. METAPROMPT surpasses the considered baselines with a significant margin of 0.11% and 2.24% on closed-set accuracy and HM scores, respectively, on average over all datasets. When, we combine our method METAPROMPT with the synthetic data generated by the Stable Diffusion process as used in ODG-CLIP, we see an increase in the overall accuracy of the closed-set and the open-set setups. This is due to the diverse generation of open and closed set samples that further helps in meta-learning of the prompts to understand the openness.

**Closed-set DG:** We report the generalization performance of our METAPROMPT for the closed-set environment in Table 2. We observe a similar trend in performance when comparing METAPROMPT against CNN-based methods: SWAD by 17.97+%, EoA by 17.17+%, DandelionNet by 16.84+%, and ODG-Net by 21.17+%. In addition, METAPROMPT surpasses zero-shot CLIP, CoOp, CoCoOp, MaPLe, LASP, STYLIP, PromptSRC, and ODG-CLIP by 17.5%, 15%, 14.5%, 12.8%, 13.4%, 10.6%, 11.9% and 0.1%, respectively, on the average over the five datasets of DomainBed benchmark. We also find that METAPROMPT outperforms the referred state-of-the-art methods on the VLCS, OfficeHome, Digit-DG, and Mini-DomainNet datasets at least by 0.18%, 0.44%, 0.18% and 0.55%, respectively. The enhanced performance in METAPROMPT can be attributed to improved visual feature extraction and our novel method of prompt learning, which promotes a more cohesive alignment between image and prompt characteristics. By simultaneously tackling both open-set and closed-set classification tasks, we enhance the discriminative nature of the embedding space. This dual focus ensures the model performs well in distinguishing known classes (closed-set) while effectively identifying and handling unknown classes (open-set). This results in a well-balanced and cohesive performance across open-set and closed-set scenarios, a synergy often absent in alternative models.

## 4.3 Ablation Studies

**Importance of different learnable components:** We ablate our proposed METAPROMPT to evaluate the effectiveness of its components: representation loss $\mathcal{L}_{Rep}$, the domain-agnostic semantic attention module ($Sem_{Att}$), and the semantic token ($T_{sem}$), as shown in Table 3. Omitting $\mathcal{L}_{Rep}$ results in an average HM score of 75.16% across five datasets. Including both $\mathcal{L}_{Rep}$ and $Sem_{Att}$, but excluding $T_{sem}$, improves the score by 2.54%. This highlights the significance of $\mathcal{L}_{Rep}$ in enhancing the model's ability to distinguish between known and unknown classes.

Table 3: Ablation of our proposed METAPROMPT with its components on the five benchmark datasets, where w/o denotes the METAPROMPT without the respective component. We report best scores in **bold**.

| Components | PACS | | VLCS | | OfficeHome | | Digits-DG | | M-Dataset | | M-DomainNet | | Average | |
|---|---|---|---|---|---|---|---|---|---|---|---|---|---|---|
| | Acc | HM | Acc | HM | Acc | HM | Acc | HM | Acc | HM | Acc | HM | Acc | HM |
| w/o $\mathcal{L}_{Rep}$ | 93.59 | 82.55 | 90.32 | 72.21 | 87.68 | 80.33 | 91.16 | 67.49 | 80.72 | 73.24 | 88.46 | 88.57 | 88.66 | 77.40 |
| w/o $T_{sem}$ | 95.20 | 84.49 | 92.27 | 75.35 | 89.94 | 83.52 | 92.35 | 68.36 | 84.41 | 76.78 | 90.28 | 90.18 | 90.74 | 79.78 |
| w/o $Sem_{Att}$ | 95.43 | 88.33 | 92.54 | 78.97 | 90.36 | 86.75 | 92.43 | 71.78 | 84.56 | 80.18 | 91.04 | 92.35 | 91.06 | 83.06 |
| with $T_{sem}$ only | 94.56 | 80.71 | 90.41 | 72.03 | 85.60 | 81.34 | 92.04 | 72.35 | 80.31 | 79.27 | 89.25 | 86.25 | 88.70 | 78.66 |
| METAPROMPT (Full) | **98.87** | **99.89** | **95.94** | **89.79** | **96.83** | **97.57** | **93.15** | **80.06** | **86.63** | **91.76** | **94.75** | **96.52** | **94.36** | **92.60** |

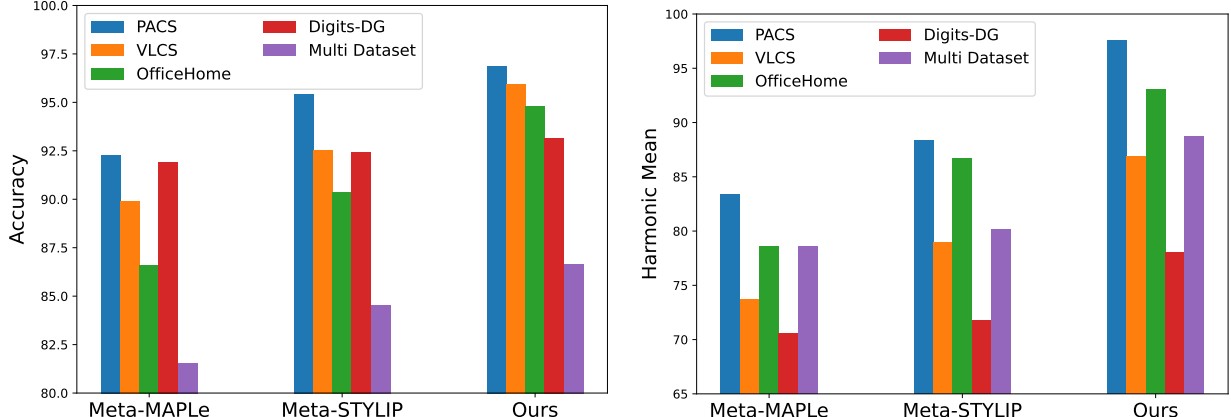

Figure 3: Performance comparison (**Left:** Accuracy, and **Right:** Harmonic Mean) of our proposed METAPROMPT with the meta-learning setup in the MAPLe Khattak et al. (2023a) and STYLIP Bose et al. (2024).

Next, we include domain-agnostic semantic information from the $Sem_{Att}$ block, resulting in a 3.5% improvement in classifying open samples compared to the previous setting. Finally, training METAPROMPT with all components results in a significant boost in open-domain generalization. We achieve the highest HM scores across all settings, with improvements of 11.6%, 11.1%, 10.8%, 8.3%, and 11.6% on the PACS, VLCS, OfficeHome, Digit-DG, and Multi-Dataset benchmarks, respectively. These results suggest that incorporating representation loss, semantic tokens, and domain-agnostic semantic information enhances the classification of in-distribution and out-of-distribution samples, effectively bridging the gap between the two.

**Comparison with prompt-learning based methods under meta-training setup:** METAPROMPT performs consistently better than MaPLe and STYLIP across the five datasets (PACS, VLCS, Office Home, Digits-DG and Multi-Dataset) as shown in Figure 3. This is possible since the SemAtt block generates the $T_{Sem}$ prompt, which encodes the global semantic information and, after being meta-trained, can determine the 'known' or 'unknown' samples. This further influences the overall latent space of the image features towards fine-grained open-set detection. Since in other methods, there are no schemes associated with the latent space of the image and the prompt embeddings, thus their performance is worse compared to our model. In Table 1, we show a complete quantitative analysis of the efficacy of our meta-training setup over the methods MaPLe, STYLIP, and ODG-CLIP. We see that when ODG-CLIP is trained in our meta-training setup, the overall closed and open set accuracy increases compared to the performance of ODG-CLIP without any meta-training. This leads to the conclusion that our meta-training setup is useful to make the learnable prompts distinguish between the close and open samples.

**Ablation with varying number of training shots:** Our proposed METAPROMPT is trained with fewer samples using a meta-learning-based approach. To evaluate the efficacy of METAPROMPT based on the number of training samples, we conduct experiments with varying numbers of training samples. In Figure 4, we report the accuracy and harmonic mean for different numbers of shots, i.e., 5, 10, 15, and 20, on the PACS, VLCS, OfficeHome, Digit-DG, and Multi-Dataset. We observe that the generalization performance increases with the number of training shots.

**Zero-shot openness performance:** In Table 4, we show the zero-shot (ZS) performance of our METAPROMPT for detecting unknown classes into clusters of their type, assuming the availability of label

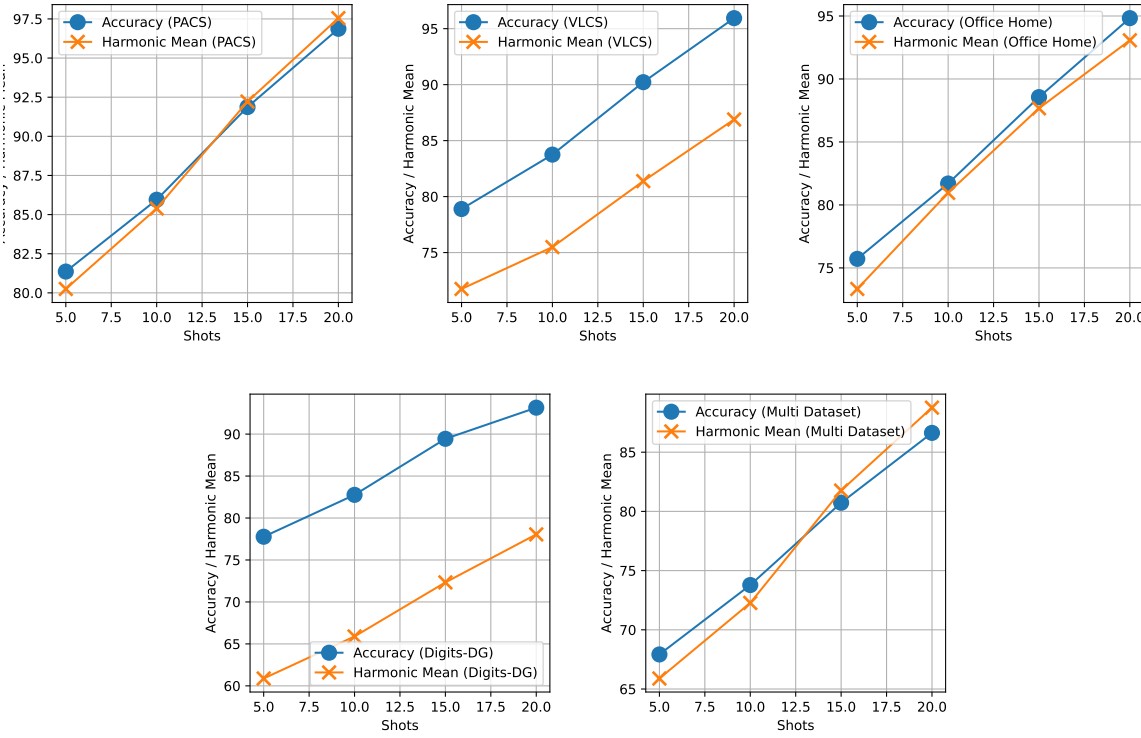

Figure 4: Ablation of our proposed METAPROMPT on varying the number of shots for the five datasets, i.e., PACS, VLCS, OfficeHome, Digit-DG, and Multi-Datasets.

information for the unknown classes. The results indicate that our method without meta-training performs poorly compared to our method with meta-training, highlighting the efficacy of the meta-learning approach in detecting and segregating unknown classes using learnable prompts. We also report the per-class open set accuracy performance of METAPROMPT without the representation loss $L_{Rep}$, observing that unknown class samples fail to form well-clustered groups in the metric space, resulting in poorer performance compared to METAPROMPT with both meta-training and representation loss.

Table 4: Ablation of our METAPROMPT for the Zero-Shot open-set accuracy of each class over the PACS, VLCS, OfficeHome, Digit-DG, and Multi-Dataset. The model is trained in two configurations: once without the Meta-Training algorithm and once without the representation loss $L_{Rep}$, denoted as w/o.

| Methods | PACS | VLCS | OfficeHome | Digits-DG | Multi-Dataset |
|---|---|---|---|---|---|
| METAPROMPT w/o meta-training | 27.66 | 19.78 | 22.67 | 17.75 | 23.67 |
| METAPROMPT w/o $L_{Rep}$ | 56.64 | 43.32 | 53.76 | 31.49 | 46.61 |
| METAPROMPT (Full) | **98.24** | **82.49** | **93.37** | **70.16** | **91.99** |

**t-SNE Visualization for open-sample clustering:** In Figure 5, we present the t-SNE visualizations of the metric space for the complete METAPROMPT compared to METAPROMPT without the representation loss $L_{Rep}$. The left panel shows the t-SNE obtained using METAPROMPT without $L_{Rep}$, while the right panel depicts the t-SNE from the complete METAPROMPT with $L_{Rep}$. These visualizations highlight the efficacy of $L_{Rep}$ in separating unknown classes into distinct clusters, thereby improving the detection of out-of-distribution samples.

**Computational Complexity:** We provide insights over the computational complexity of our proposed METAPROMPT method. We have demonstrated the efficacy of METAPROMPT in comparison to state-of-

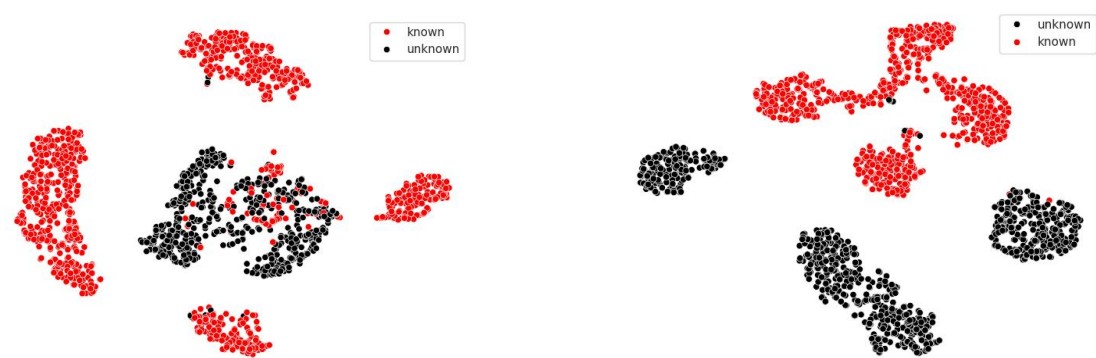

Figure 5: Comparing the t-SNE visualization obtained from our METAPROMPT under the setting of without (**left**) and with (**right**) use of the representation losses defined in Equation 5

the-art ODG methods. Additionally, we provide an analysis of the computational overhead inherent in our METAPROMPT approach for the OfficeHome dataset in the Table 5 below. This shows METAPROMPT have the least number of parameters with the lowest GFLOPS compared to the current state-of-art methods. Thus reducing the overall training time and testing time.

| Method | Parameters (#M) | GFLOPs | Training time (min) | Testing time (min) |
|---|---|---|---|---|
| ODG-CLIP | 142 | 253.6 | 65 | 1.3 |
| Meta-ODG-CLIP | 156 | 255.4 | 70 | 1.3 |
| Ours | 12 | 47.5 | 27 | 0.6 |

Table 5: Comparison of model methods in terms of parameters, GFLOPs, training time, and testing time.

## 5 Conclusion and Future Scopes

This paper presents a novel meta-learning-based open-domain generalization method named METAPROMPT. We leverage the pre-trained CLIP model and perform prompt learning constrained on the input image to detect out-of-distribution instances. In METAPROMPT , we extract features from the intermediate layers of CLIP's image encoder to learn BatchNorm statistics, aiding in the adaptation to unseen target domains by learning domain-agnostic features. Additionally, we obtain semantic attention tokens from these intermediate layers to generate semantic-aware context tokens for text embeddings. These tokens are then concatenated with the final image features from CLIP for contrastive training with the text embeddings. Combined with meta-learning, this image-conditioned, semantic-aware prompt learning setup excels in generalizing open-set classification for diverse unseen target distributions. Empirically, we report results for both open-set and closed-set DG scenarios with six benchmark datasets. Our proposed METAPROMPT surpasses all the state-of-the-art methods by a significant margin. In future work, we aim to extend our approach to challenging tasks such as open-set domain generalization for dense prediction and object detection.

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
