# OpenReview forum: "Meta-Learning to Teach Semantic Prompts for Open Domain Generalization in Vision-Language Models"
_TMLR — Accepted by TMLR_

### Review · Reviewer_NaUB · 2024-09-25

**Summary Of Contributions:**

In this paper,  the author  introduced  a novel meta-learning-based prompt learning method, MetaPrompt, for open domain
generalization;  They  leveraged the multi-scale image features of the pre-trained CLIP to generate the BatchNorm statistics.
Additionally,  they proposed  the semantic attention block (SemAtt) that helps generate more  semantically
meaningful prompt tokens and improves clustering between open-set and closed-set classes.

**Audience:**

Yes

**Claims And Evidence:**

Yes

**Requested Changes:**

The author's response resolved my issue. I agree to receive this paper.

**Strengths And Weaknesses:**

The paper  introduces three key innovations: First, they  approach ODG as a multi-class classification problem
that includes both known and novel categories, designing novel prompts capable of detecting unknown class samples across multiple domains. These prompts are trained using Meta-Learning with momentum updates, enabling smooth and accurate differentiation between known and unknown classes. Second, they  introduce a novel domain-agnostic semantic attention-based prompt alongside domain-focused prompts to enhance robustness in classifying unknown classes across various domains. Finally, they  incorporate an unsupervised contrastive loss during episodic Meta-Training, which reinforces the boundaries in the metric space between known and unknown classes, thereby enhancing “unknown” class awareness in the prompts.  MetaPrompt has demonstrated its superiority through extensive testing
on diverse datasets, excelling in both closed and open-set DG scenarios and consistently outperforming existing solutions.

The author's response resolved my issue. I agree to receive this paper.

---

> ### Author Response · Authors · 2024-10-29
> **Rebuttal for reviewer NaUB**
>
> We thank Reviewer NaUB for the insightful comments. We have provided the responses of your concerns below.
>
> $$\textbf{1. Universality of our proposed method}$$
> We thank you for providing this suggestion to check the universality of our method. We apply our meta-learning approach three of the prominent VLM baselines MaPLe, StyLIP and ODG-CLIP. In the Figure 3 of our manuscript we had already presented the efficacy of our meta-learning algorithm when applied over the said baseline methods MaPLe and StyLIP. The table is given below presents a precise quantitative values over all the methods as follows,
>
>         Ablation	             PACS           VLCS          OfficeHome	    DigitDG	MultiDataset    MiniDomainNet        Average
>                            Acc    H-Score	Acc	H-Score  Acc	H-Score	 Acc	H-Score	 Acc	H-Score	 Acc	H-Score	 Acc	H-Score
> 		ODG-CLIP		99.53	99.70	95.71	86.53	98.32	96.08	91.53	78.27	86.40	90.00	95.68	94.48     94.23 90.84
> 		Meta-ODG-CLIP       99.78	99.95	96.52	88.44	98.66	98.29	92.94	80.16	86.51	92.33	96.43	95.58     95.14 92.45
>         Meta-MaPLe          92.35	84.44	89.42	73.56	86.33	78.61	91.23	71.69	81.79	78.75	79.48	69.81    86.77 76.14
>         Meta-StyLIP         95.67	88.39	93.25	78.87	91.57	87.35	92.18	72.73	84.23	83.44	86.77	78.08    90.61 81.47
>         Ours                98.87	99.89	95.94	89.79	96.83	97.57	93.15	80.06	86.63	91.76	94.75	96.52    94.36 92.60
>
> Here Meta-MaPLe, Meta-StyLIP and Meta-ODG-CLIP refer to the meta-leraning version of MaPLe, StyLIP and ODG-CLIP. The results clearly show that both of the closed-set performances and harmonic scores of the models get improved while adding the meta-learning paradigm in the ODG task.
> We also update the manuscript with the suggested changes.
>
> $$\textbf{2. Computational overhead.}$$
> We thank reviewer for their valuable insights on the computational complexities in our proposed MetaPrompt method. We have demonstrated the efficacy of MetaPrompt in comparison to state-of-the-art ODG methods. Additionally, we provide an analysis of the computational overhead inherent in our MetaPrompt approach for the OfficeHome dataset in the table below.
>
>     Method       Parameters (#M)   GFLOPs Training time (min) Testing time (min)
>     ODG-CLIP         142          253.6             65           1.3
>     Meta-ODG-CLIP    156          255.4             70           1.3
>     Ours             12          47.5             27           0.6
>
> $$\textbf{3. Limitations of MetaPrompt}$$
>  The ODG set up is not yet explored for dense prediction tasks. Semantic segmentation or depth estimation, models must make predictions at the pixel level, which requires a nuanced understanding of spatial relationships and contextual information within the image. When faced with unseen domain characteristics with novel categories, such as different lighting conditions, object appearances, or background clutter, the model's ability to interpret these spatial cues can be severely compromised. In addition, ODG methods often rely on feature extraction and adaptation techniques that may not effectively capture the fine-grained details necessarily for unseen/novel categories in the case of accurate classifications or dense predictions.

---

> > ### Comment · Reviewer_NaUB · 2025-02-09
> > **Meta-Learning to Teach Semantic Prompts for Open Domain Generalization in Vision-Language Models**
> >
> > The paper introduces three key innovations: First, they approach ODG as a multi-class classification problem that includes both known and novel categories, designing novel prompts capable of detecting unknown class samples across multiple domains. These prompts are trained using Meta-Learning with momentum updates, enabling smooth and accurate differentiation between known and unknown classes. Second, they introduce a novel domain-agnostic semantic attention-based prompt alongside domain-focused prompts to enhance robustness in classifying unknown classes across various domains. Finally, they incorporate an unsupervised contrastive loss during episodic Meta-Training, which reinforces the boundaries in the metric space between known and unknown classes, thereby enhancing “unknown” class awareness in the prompts. MetaPrompt has demonstrated its superiority through extensive testing on diverse datasets, excelling in both closed and open-set DG scenarios and consistently outperforming existing solutions.
> >
> > The author's response resolved my issue. I agree to receive this paper.

---

### Review · Reviewer_j971 · 2024-10-14

**Summary Of Contributions:**

The authors present a meta-learning-based approach to address open domain generalization (ODG), building on the strengths of CLIP. This method shares some components with ODG-CLIP [1], such as the use of "unknown class" labels and prompt learning. It differentiates with ODG-CLIP in three key points: (1) it employs a different architectural design, (2) introduces a modified loss function (representation loss), and (3) utilizes meta-learning for training, while ODG-CLIP relies on synthetic sample generation through a diffusion model.

[1] Mainak Singha, Ankit Jha, Shirsha Bose, Ashwin Nair, Moloud Abdar, Biplab Banerjee. Unknown Prompt, the only Lacuna: Unveiling CLIP's Potential for Open Domain Generalization. CVPR 2024.

**Audience:**

Yes

**Broader Impact Concerns:**

I do not have any concerns regarding Broader Impact Concerns.

**Claims And Evidence:**

Yes

**Requested Changes:**

See the weakness above.

**Strengths And Weaknesses:**

**Strength**
- The paper provides a good walkthrough of the components of its proposed method.
- The ablation studies regarding each of these components are provided.

**Weakness**

I have concerns about the clarity in demonstrating where the improvements over ODG-CLIP truly come from. Specifically:

- The performance gains compared to ODG-CLIP is minimal. For instance, ODG-CLIP reported that their accuracy of 94.23±0.19 for Table 1 setting, while this method (MetaPrompt) achives 94.36, which lacks statistical significance. Similarly, closed set generalization performance is also incremental, only showing 0.14 improvement compared to ODG-CLIP.
- Not significantly outperforming SOTA method is not necessarily a critical issue if the method has advantages in other aspects such as data efficiency or training efficiency, which I do not see any of them. Maybe the authors could elaborate how efficient this method is compared to ODG-CLIP.
- I would like to see how the performance changes compared to ODG-CLIP, isolating the two components that author introduces, which are (1) different architecture/loss design and (2) training method (meta-learning). There does not seem to be a clear rationale for why the proposed loss (representation loss) is particularly well-suited for a meta-learning framework, as it could be optimized by using synthetic data for unknown samples as in ODG-CLIP. I would like to know if these two components equally contribute in improving the performance of ODG-CLIP. Specifically, I would like to see the performance of

    - Proposed method (MetaPrompt)'s architecture/loss + learning with synthetic data as in ODG-CLIP.
    - Proposed method (MetaPrompt)'s training method (meta-learning) + ODG-CLIP's architecture/loss.

---

> ### Author Response · Authors · 2024-10-29
> **Rebuttal for reviewer j971**
>
> We thank Reviewer j971 for the insightful comments. We have provided the responses of your concerns below.
>
> $$\textbf{1. Minimal performance gain over ODG-CLIP.}$$
> We agree that MetaPrompt has achieved minimal improvement on average closed-set accuracy compared to ODG-CLIP. To be noted, MetaPrompt is designed in a meta-training set up to learn the unknown classes on a particular episode, where the same classes could be treated as seen classes in another episode. So during inference, the model should adopt the novel classes as unknown, while preserving the closed-set performance as well. Following this, we achieve 2\% of improvement compared of ODG-CLIP, in terms of harmonic mean of closed and open-set accuracies, which depicts the better generalizability of MetaPrompt.
>
> $$\textbf{2. Efficiency of MetaPrompt compared to ODG-CLIP.}$$
> The problems we observe in ODG-CLIP are, it is highly dependent on the pre-trained Stable Diffusion model for the synthetic data generation, that becomes computationally expensive while training. The generated data may not be consistent during each training, while keeping all the hyperparameters same. However, though the data generation module will not be attached with the training process, the model could lack from vague semantic information of the generated "unknown" samples, as the "unknown" class generation has no boundary to define the desired output that could lead of hallucinated randomness. In contrast, MetaPrompt has followed a meta-learning set up, where few seen classes are considered as known classes and rest of the seen classes ae treated as unknown class in each episodes. This process reduces the necessity of novel category supervision in the model, and also mitigates the computational costs of training. In addition, the episodic unknown samples carry a meaningful semantic information to learn the `unknown' word in the CLIP space to do a better alignment between visual and text modalities.
>
>
> $$\textbf{3. Experiments of (i) MetaPrompt + Synthetic data (Stable diffusion).}$$
> We have updated the experiments in the revised manuscript.
>
>         Ablation	  	            PACS           VLCS            OfficeHome	    DigitDG	MultiDataset      MiniDomainNet     Average
>                                       Acc  H-Score	 Acc	H-Score  Acc	H-Score	 Acc	H-Score	 Acc	H-Score	 Acc	H-Score   Acc	H-Score
> 		MetaPrompt + StableDiffusion  99.13	99.95	96.53	92.85	97.58	98.89	95.27	83.69	88.65	94.14	96.19	98.79   95.56  94.72
> 		MetaPrompt (ours)		  98.87	99.89	95.94	89.79	96.83	97.57	93.15	80.06	86.63	91.76	94.75	96.52   94.36  92.60
>
> $$\textbf{4. Experiments of (ii) Meta-learning + ODG-CLIP.}$$
> We have updated the experiments in the revised manuscript.
>
>         Ablation	             PACS           VLCS          OfficeHome	    DigitDG	  MultiDataset    MiniDomainNet         Average
>                             Acc    H-Score	Acc	H-Score  Acc	H-Score	 Acc	H-Score	 Acc	H-Score	 Acc	H-Score	     Acc	H-Score
> 		ODG-CLIP		99.53	99.70	95.71	86.53	98.32	96.08	91.53	78.27	86.40	90.00	95.68	94.48       94.23 90.84
> 		MetaPrompt+ODG-CLIP 99.78	99.95	96.52	88.44	98.66	98.29	92.94	80.16	86.51	92.33	96.43	95.58       95.14 92.45
> 		Ours                98.87	99.89	95.94	89.79	96.83	97.57	93.15	80.06	86.63	91.76	94.75	96.52       94.36 92.60
>
> The synthetic data  has been generated in the same manner as mentioned in the ODG-CLIP.

---

### Review · Reviewer_sRXY · 2024-12-11

**Summary Of Contributions:**

This paper introduces MetaPrompt, a novel meta-learning approach for Open Domain Generalization (ODG) using the vision-language model CLIP. It optimizes prompts in the text encoder through meta-learning to enhance generalization across domain and category shifts. The main contributions are:

1. Innovative use of contrastive loss to align text and vision encoders and to cluster vision embeddings of similar classes.
2. Proposal of proxy text prompts that utilize visual information via semantic attention and batch normalization (BN) statistics, eliminating the need for paired vision-language datasets.
3. Formulation of ODG as a meta-learning problem to generalize to unseen classes and domains during testing.
4. Extensive evaluation of the proposed method on both open-set and closed-set domain generalization across six benchmark datasets, accompanied by comparisons with relevant baselines and ablations.

**Audience:**

Yes

**Claims And Evidence:**

Yes

**Requested Changes:**

1. Provide explicit justifications or references for the assumptions in Weakness Point 1.
2. Conduct ablation studies to substantiate the design choices mentioned in Weakness Point 3.
3. Address the clarifications in Weakness Point 4.

**Strengths And Weaknesses:**

### Strengths
1. The innovative framing of ODG as a meta-learning problem that generates optimal prompts using visual embeddings for distinguishing known and unknown classes.
2. Achievement of state-of-the-art (SOTA) performance on both open-set and closed-set DG tasks across six benchmarks, with robust comparison to prior methods.
3. Clear presentation of methods, figures, and results, making the paper accessible and easy to follow.

### Weaknesses

1. Certain assumptions are presented as axiomatic without adequate justification or references, including:
   - The claim that BN statistics capture style and content information (Section 3.2).
   - The assumption that context tokens are inherently domain-agnostic (Section 3.2).

2. Ambiguity regarding the semantic attention module:
   - Why is it described as domain-agnostic?
   - The use of "attention" is unclear, as no explicit attention mechanism appears in the module design.

3. Missing ablations for critical design components:
   - Relevance of multi-scale features shown in Figure 2. How does performance change without them?
   - Contribution of context tokens. What happens if only `Tsem` and the `CLS` token are input to the text encoder?

4. Various clarifications and typos require attention:
   - The meaning of Ω in Equation 1 and z in Equation 5 is not defined.
   - Figure 2’s text mentions upsampling for dimension matching, but the figure illustrates only downsampling layers.
   - Page 11, line 4: Improvements of 11.6%, 11.1%, etc., lack clarity regarding the baseline for comparison.

---

> ### Author Response · Authors · 2025-01-01
> **Rebuttal for reviewer sRXY**
>
> Thank You for valuable feedback
>
> $$\textbf{1. The claim that BN statistics capture style and content information:}$$
> We inspired from the following literature for the motivation of considering BN statistics for capturing style and content information  “Demystifying Neural Style Transfer” [1] and "StyLIP" [2].
>
> [1] Demystifying neural style transfer, Yanghao Li, Naiyan Wang, Jiaying Liu, Xiaodi Hou, IJCAI 2017.
>
> [2] StyLIP: Multi-Scale Style-Conditioned Prompt Learning for CLIP-based Domain Generalization, Shirsha Bose, Ankit Jha, Enrico Fini, Mainak Singha, Elisa Ricci, Biplab Banerjee, WACV 2024.
>
> $$\textbf{2. The assumption that context tokens are inherently domain-agnostic. Why is it described as domain-agnostic? }$$
> Only the context tokens ($T_{sem}$) from the output of $SemAtt$ module are domain-agnostic (we describe the reason for calling them domain-agnostic in weakness 2), not the context tokens $C_m$ from BNS module. We have corrected the mistake for mentioning $C_m$ as domain-agnostic in the line below eq. 2 in page 5.
> The semantic attention block $SemAtt$ takes the intermediate features of the pre-trained image encoder of CLIP as input, which generally provides the low-level basic visual knowledge (from the lower level layers) and task-specific knowledge (from the higher level layers). As these features do not provide any kind of domain or distribution information, we call them domain-agnostic.
>
> $$\textbf{3. The use of ``attention'' is unclear: }$$
> We apologize for the confusion. We represent the generation of semantic attention tokens ($T_{Sem}$) using the $SemAtt$ block, which has been shown in Figure 3 in the main manuscript. This block helps in the generation of semantically focus tokens which are agnostic towards diverse domains during training. During the inference, this $T_{Sem}$ tokens effectively recognize new classes, shown in Table 3.
>
> $$\textbf{4. Relevance of multi-scale features, in Figure 2 and impact on performance?}$$
> The relevance of the multi-scale features is to capture the semantic information that is distinct from the latent space, to compute the class-wise representation loss, and to generate the T_sem token. The T_sem token contains the image-related semantic information which makes the prompt-learning more robust towards identifying a novel object. Finally, we add the multi-scale features to the latent space to make the latent space aware of the distinctiveness of the multiple-class objects. We show the experiments where we remove the T_sem token to show the efficacy of this image-information-related token.
> The CLS token represents the class name, e.g.,” cat,” “dog,” etc., when we use them for training or testing. So, this is a non-learnable token; it is just the class text names.
> If we use the T_sem and CLS token then the overall concept of making the domain generalization will fail and it will be biased towards the objectness of the training samples. This will collapse the idea of making the prompts more conscious towards the domain-generalized novel object identifying, the prompt-tokens are taken from the different depths of the image encoder and they contain information of the batch norm features, unlike the T_sem which comes from the convolution layer which is devoid of any domain related information.
>
> $$\textbf{5. Contribution of context tokens. Using $T_{sem}$ and $CLS$ token as the input.}$$
> Without the multi-scale features we would not have the $T_{sem}$ token as well as the Representation Loss $L_{rep}$ thus reducing the overall accuracy of novel class and known class.
>
>             Ablation	              PACS           VLCS          OfficeHome	    DigitDG	  MultiDataset    MiniDomainNet         Average
>                                      Acc    H-Score	  Acc	H-Score  Acc	H-Score	 Acc	H-Score	 Acc	H-Score	 Acc	H-Score	  Acc H-Score
>           Without $T_{Sem}$       95.20	84.49	92.27	75.35	89.94	83.52	92.35	68.36	84.41	76.78	90.28	90.18	90.74	79.78
> 	     With $T_{Sem}$ only      94.56	80.71	90.41	72.03	85.60	81.34	92.04	72.35	80.31	79.27	89.25	86.25	88.70	78.66
> 	       MetaPROMPT (Ours)      98.87	99.89	95.94	89.79	96.83	97.57	93.15	80.06	86.63	91.76	94.75	96.52	94.36	92.60
>
> $$\textbf{7.The meaning of Ω in Equation 1 and z in Equation 5 is not defined. Figure 2’s text mentions upsampling for dimension matching, but the figure illustrates only downsampling layers. Page 11, line 4: Improvements of 11.6%, 11.1%, etc., lack clarity regarding the baseline for comparison. }$$
> In the paper, we compared our MetaPROMPT against CLIP-based baselines such as ODG-CLIP and CLIPN, shown in Table 1 and Table 2. Along with this, we showcase our proposed method with respect to the Meta-learning based approaches, shown in Table 4 of main manuscript. The experimental results signify the outperformance of our MetaPROMPT against all the state-of-the-art and baseline settings.
> We have updated the Figure 2 and notations used in Equation 1 and Equation 5 in the revised manuscript.

---

### Decision · Action_Editor_9mNd · 2025-03-11

**Recommendation:** Accept as is

**Comment:**

The paper presents MetaPrompt to improve Open Domain Generalization by combining CLIP’s vision-language strengths with Meta-Learning for better unknown class detection. It introduces novel prompts, a domain-agnostic semantic attention mechanism, and contrastive loss to enhance robustness across domain and category shifts, outperforming existing methods.

The reviewers highlight the core strengths as
- the contrastive loss for improving vision-text alignment and cluster similar vision embeddings
- proxy text prompts that utilize semantic attention and BN statistics, eliminating the need for paired vision-language datasets,
- Casting ODG as a meta-learning problem enhances generalization of unseen classes and domains.

The reviewers cited extensive evaluations of six benchmark datasets, including comparisons and ablation studies, which validated its effectiveness.

In the first round of review, however, reviewers share some concerns regarding
- inadequate justification in Section 3.2 and
- missing ablations for critical design components
- method exposition issues.

The authors' revision effectively addressed these raised concerns. After the discussions, two of the reviewers are leaning toward acceptance. One reviewer did not respond with a final decision, but based on the comments, the concerns have been resolved. The AE agrees with the reviewers and recommends acceptance.

**Audience:**

Yes, the audience will be interested in the findings of this paper.

**Claims And Evidence:**

Yes, the claims in the submission have been sufficiently supported by evidence. The additional ablation after the revision further validates the semantic token's effectiveness (Table 3).